# Development of a Microfluidic Device for CD4^+^ T Cell Isolation and Automated Enumeration from Whole Blood

**DOI:** 10.3390/bios12010012

**Published:** 2021-12-28

**Authors:** Robert D. Fennell, Mazhar Sher, Waseem Asghar

**Affiliations:** 1Asghar-Lab, Micro and Nanotechnology in Medicine, College of Engineering and Computer Science, Boca Raton, FL 33431, USA; rfennel1@my.fau.edu (R.D.F.); msher2015@fau.edu (M.S.); 2Department of Electrical Engineering and Computer Science, Florida Atlantic University, Boca Raton, FL 33431, USA; 3Department of Biological Sciences (Courtesy Appointment), Florida Atlantic University, Boca Raton, FL 33431, USA

**Keywords:** CD4^+^ T helper cells, microfluidic chip, microbeads, wide-field optical system, ImageJ

## Abstract

The development of point-of-care, cost-effective, and easy-to-use assays for the accurate counting of CD4^+^ T cells remains an important focus for HIV-1 disease management. The CD4^+^ T cell count provides an indication regarding the overall success of HIV-1 treatments. The CD4^+^ T count information is equally important for both resource-constrained regions and areas with extensive resources. Hospitals and other allied facilities may be overwhelmed by epidemics or other disasters. An assay for a physician’s office or other home-based setting is becoming increasingly popular. We have developed a technology for the rapid quantification of CD4^+^ T cells. A double antibody selection process, utilizing anti-CD4 and anti-CD3 antibodies, is tested and provides a high specificity. The assay utilizes a microfluidic chip coated with the anti-CD3 antibody, having an improved antibody avidity. As a result of enhanced binding, a higher flow rate can be applied that enables an improved channel washing to reduce non-specific bindings. A wide-field optical imaging system is also developed that provides the rapid quantification of cells. The designed optical setup is portable and low-cost. An ImageJ-based program is developed for the automatic counting of CD4^+^ T cells. We have successfully isolated and counted CD4^+^ T cells with high specificity and efficiency greater than 90%.

## 1. Introduction

There is a need to develop accurate cell quantification assays to achieve early-stage disease detection, treatment, and monitoring. Various cell quantification assays have been developed, tested, and validated for a multitude of diseases over the course of time [1]. The Coulter Principle has resulted in the Coulter counters being able to measure cell size and impedance in an electrolyte solution [2,3]. The principle has been extended to interactions with light as well. As a result of these advances, several sophisticated and advanced laboratory-based devices have been tested and approved for accurate cell quantification purposes. Such devices require well-trained personnel in well-equipped laboratories. Resource-limited settings lack these facilities. Hence, there is an unmet need to develop cost-effective, easy-to-use, and rapid disease diagnostic devices at the point-of-care settings, POC. The World Health Organization (WHO) has set these guidelines for future diagnostic equipment using the acronym ASSURED, which stands for affordable, sensitive, specific, user-friendly, rapid and robust, equipment-free, and deliverable. Devices developed based on these guidelines would be equally beneficial for both resource-enabled and resource-limited countries. This lack of adequate resources also applies to physicians’ offices, patients’ homes, and rapidly growing telemedicine situations. Due to the need for speed and on-site diagnosis, the ‘Sample in, Answer-out’ type of assays is gaining popularity.

Early disease diagnosis is a critical factor, especially in outbreaks of infectious diseases, such as HIV (human immunodeficiency virus), Ebola, Zika, and SARS-CoV-2 [4,5,6]. Urgent and timely clinical decisions can help to detect and curtail the spread of infectious diseases. Sending samples and receiving their results from a clinical laboratory often takes days. Higher throughput and rapid assays are the need of the day. Currently, hospitals often create their own testing facilities to reduce the turnaround time for the same-hour diagnosis. The development of POC diagnostic tools would help to make a patient’s bedside testing possible. The test results could be obtained within the least amount of time, which enables the physician to make an early clinical decision and explore further options for treatment.

POC devices are supposed to be portable, cost-effective, and environment-friendly [7,8,9]. It is estimated that the biosensor market will expand further in the coming years. The current developments in cellular communications, smartphone imaging systems, integrated circuit technology, along with disposable microfluidic devices, can be utilized for the future POC devices in resource-limited areas.

The CD4^+^ T count provides important information about the overall success of HIV treatment. Once HIV is diagnosed, the treatment is evaluated by the CD4^+^ T lymphocyte cell count and CD4/CD8 ratios. As the disease is treated, several assays are required. Flow cytometry is a reliable and accurate method for the quantification of CD4 cells, but it has high equipment and test costs and requires skilled resources for operation, results analyses, and maintenance. There is a dire need to develop microchip-based assays for the enumeration of CD4^+^ T cells. The main task is to isolate and quantify CD4^+^ T cells from a drop of blood. This whole process could lead to a POC assay to be used in medically resource-poor locations that cannot afford expensive diagnostic testing.

The need for microfluidic devices has been explored by a number of researchers [8,10,11]. Earlier, researchers developed a microfluidic device for the label-free CD4^+^ T cell counting of HIV-infected subjects [11]. They explored the use of a microfluidic module coated with a specific antibody for the isolation of CD4^+^ T cells and monocytes. A well-controlled fluid flow was used to separate monocytes from CD4^+^ T lymphocytes. A consistent well-controlled flow is difficult to control in a field location inside the microfluidic channel. Therefore, there is the possibility of an adverse effect on the captured CD4^+^ T lymphocytes, and, as a result, some of the cells may be forced to detach from the microchip along with the monocytes. In one other study, Moon et.al. have developed a portable microfluidic platform to count the CD4^+^ T cells of HIV-infected patients [12]. A microchip was coated with anti-CD4 antibody to capture the target CD4^+^ T cells from a fingerpick volume of whole blood. The captured cells were imaged using the lensless CCD platform. Their setup requires an automated cell counting program developed in MATLAB to identify the CD4^+^ T lymphocyte cells from monocytes.

Here, we present an alternate approach to identify CD4^+^ T cell lymphocytes from monocytes. We describe an improved microfluidic chip process that separates a sparse number of CD4^+^ T lymphocytes from whole blood samples. The developed method utilizes anti-CD4 antibody-conjugated magnetic beads to capture the CD4^+^ T cells from a drop of whole blood with high specificity. The CD4^+^ T lymphocytes are separated from monocytes with an anti-CD3 antibody-coated microchip. A washing step is then performed to get rid of the unneeded whole blood components and unattached magnetic beads from the microfluidic chip. The images of the captured CD4^+^ T cells are then recorded with the help of a custom-built wide-field optical setup. The developed wide-field optical setup helps to cover a large field of view, enabling a sufficient sample size. The images are analyzed using a computer program we developed with Image J, and the cells are counted automatically.

The developed method has several advantages over the existing microfluidic techniques. Here, the avidity improvement is caused by the utilization of an anti-CD3 antibody for coating the bottom glass substrate of the microfluidic chip along with the anti-CD4 antibody coated magnetic beads. This approach has been applied for the first time, to the best of our knowledge, in microfluidic-based CD4^+^ T cells quantification assays. Along with the enhanced antibody avidity, this approach also enabled much better CD4^+^ T cell isolations. Initially, we were able to isolate all the CD4^+^ T cells from the blood samples using the anti-CD4 coated antibody. Then, the monocytes were separated from those initially isolated CD4^+^ T cells using the anti-CD3 coated to the glass slide inside a microfluidic chip. We have found that the CD4^+^ T cells separation can be efficiently quantified using this approach with less shear stress on the cell.

## 2. Materials and Methods

### 2.1. Fabrication of the Microchip Module

The complete process list, the material source list, and a process datasheet for the microchip fabrication are presented in detail in the Appendix A. The following describes the reagents, microchip design, the process design and development, the optical design and development, followed by the results obtained.

### 2.2. The Reagents

A complete list of materials and reagents is presented in the Appendix A. Anti-CD3 antibody, UCHT1(biotin), part number ab191112 was obtained from Abcam, and it was used to coat the bottom glass substrate of the microchip. Dynabeads, part number 1145D, were obtained from ThermoFisher (Waltham, MA, USA); they were precoated with a proprietary anti-CD4 antibody.

To avoid bead clumping and unwanted adhesion to the module channel, we used ultra-pure Dulbecco’s phosphate-buffered saline solution (DPBS) throughout the washing step. EDTA: ethylenediaminetetraacetic acid was used as an anticoagulant.

PMMA, a polymethylmethacrylate sheet, was utilized as the base material for making microfluidic chips. PMMA can easily be cut with a laser cutter. Various PMMA layers were joined together with the help of double-sided adhesive tape (DSA) to form a composite microfluidic chip.

The bottom layer of a microfluidic device was chemically modified to enable antibody attachment. The in-house processing uses 3-MPS (3-Mercaptopropyl) trimethoxysilane (3-MPS, CN: 175617) and cross-linker GMBS: N-γ-Maleimidobutyryloxy succinimide ester. GMBS is a crosslinker from the amino and sulfhydryl groups. It comes as a solid and is water insoluble. Therefore, it must be mixed with DMSO or DMF [3]. We used DMSO to dissolve the GMBS so that it could be applied to the substrate.

### 2.3. The Microfluidic Design

A drawing of the microchip module and its photograph are shown in Figure 1. The modules were designed using AutoCAD software. The AutoCAD software was linked to a laser cutter VLS 2.30 laser cutter (VersaLaser, Scottsdale, AZ, USA). PMMA and DSA sheets were cut according to the design requirements as per previously developed method [13,14,15,16]. The dimensions of microfluidic channels were selected in such a manner that there were three microchannels in each chip for running two samples and one control.

### 2.4. Immobilization of Antibody to Glass Substrate

Streptavidin-coated glass slides were prepared in-house. Figure 2 presents the basic process for microchip fabrication and was previously demonstrated [7,10,17]. Briefly, a biotinylated anti-CD3 antibody was applied to the substrate. The anti-CD3, from Abcam, (Catalog No. AB191112) recognizes the CD3 antigen of the TCR/CD3 complex on mature human T cells. The antibody reacts with the epsilon chain of the CD3 complex. The developed assay utilizes the stronger avidity of anti-CD3 antibody coated on a microfluidic chip. More explanation on stronger avidity is given in the Appendix A.

Our initial process used anti-CD4 antibodies (ab28069) on the substrate and anti-CD3 antibodies on the microspheres. We encountered difficulties in the washing process. In order to clear out the red blood cells, we need at least 50 μL/min flow rate inside microfluidic channels. However, anything greater than 25 μL/min swept out some of the white blood cells also. We reversed the antibodies. This created an improved binding strength to capture CD4^+^ T helper cells to the substrate. We were able to use 50 μL/min cleaning flow with good results. Better adhesion to the prepared substrate was achieved as the Streptavidin layer was already well adhered, enabling a more vigorous wash cycle.

To find out the capture efficiency, the Dynabeads standard magnetic separation method was used to separate out CD4^+^ T cells along with the Dynabeads in a tube. Dynabeads were functionalized by a custom, proprietary anti-CD4 antibody. Initially, Dynabeads were mixed with the blood sample in a tube. Then, the isolation of CD4^+^ T cells was performed by an external magnet; the unbound entities from the blood were removed with the help of a pipette washing. The DPBS was added to the captured beads-cell complexes, and the washing step was repeated again. This second washing step was performed to get rid of any non-desired entity. Hence, CD4^+^ T cells and beads remained inside the tube. An HC count was then used to count the cells under an inverted microscope (Nikon Eclipse TE2000-S; Nikon, Japan) at 10× [18]. A standard counting method was used. The four corner sections were each counted for the cells of interest, and then the cell count was averaged. The cell count was then multiplied by the dilution factor.

For comparison, the microfluidic chip was also used for capturing CD4^+^ T cells. The number of CD4^+^ T cells isolated with the help of Dynabeads was confirmed using an alternate commercially available kit, “EasySep” from Stemcell Technologies [19], which isolated CD4^+^ T by a negative selection. These cells were then used as a validation check in one of the parallel channels in the microfluidic device to verify the process. As an additional test to confirm white cell identification, NucBlue, a cell permanent counterstain, was used with a Fluorescent Cell Imager (ZOE, Bio-Rad Laboratories, Hercules, CA, USA) with 20× objective lens cell imaging system.

The Dynabeads and CD4^+^ T cells were injected into the microchip having the anti-CD3 antibody functionalization. Figure 3 portrays a full wide-field view, taken on our lensless equipment, that shows the full view of the microfluidic chip channel.

DPBS was used to wash the channels that reduced the clumping of beads and cells. The channel washing was performed at a high flow rate of 50 μL/minute for 8 min. There were only a few beads not attached to CD4^+^ T cells that were left in the channel.

### 2.5. Measurement of Capture Efficiency

As previously mentioned, the cell isolation process used the double selection method with anti-CD4 antibody-coated Dynabeads and an anti-CD3 antibody immobilized to the glass slide. The images of the captured cells were taken using the microscope and are called the microscope count, MC. The image areas, microchip channel volume, and dilution amount are taken into account. Multiple images are necessary because the inverted microscope (Nikon Eclipse TE2000-S) trades off magnification for coverage area, and the CD4^+^ T count can be low per unit area. A wide area imaging technique was developed in earlier work [20] to compensate for this, and the design is described in Section 3 D. Capture efficiency was then the module microscope count (MC) divided by HC, the hemocytometer count. The module count, MC, was done by stitching together 8 photographs down a channel and then counting the CD4^+^ T cells. In both cases, at this magnification, resolution, and narrow field of view, the low number of cells visible per image requires several different pictures to get a sufficient sample. This is also true for a hemocytometer, HC, where several photos were taken over several sectors. The images were then reviewed and CD4^+^ T cells were identified and counted per unit volume.

MC: count on inverted microscope (Nikon Eclipse TE2000-S) of the microchip module.

HC: count on inverted microscope (Nikon Eclipse TE2000-S) of the hemocytometer, Dynabeads process.
Efficiency = MC/HC %(1)

### 2.6. Measurement of Specificity

To verify our process separation of CD4^+^ T cells, a cell permeant counterstain was used, NucBlue, (Hoechst 33342), that emits blue fluorescent light at 460 nm when bound to DNA and excited by UV light. A Fluorescent Cell Imager (ZOE, Bio-Rad Laboratories, Hercules, CA, USA) with 20× objective lens was used. Brightfield (BF) and fluorescent images (FC) were created, stored, and merged, as shown in Appendix A. An ImageJ program was then used to count cells and compare the BF and FC count. Visual confirmation was done with the merged view. Specificity was then the FC divided by BF.
Specificity = FC/BF %(2)

### 2.7. Development of a Lensless System to Enable Wide-Field-Imaging

In the development of a lensless imaging system, the key design issue is to generate a near monochromatic light source to ensure a parallel source without scattering. The module needs to be as close as possible to the CCD imaging device. An experimental lab-quality device was built as shown in Figure 3. It has the advantage of an X, Y, Z incremental movement of the microchip module. Multiple channels can easily be observed and brought into view. A portable lensless device is shown in Figure 3 that is suitable for a single channel micromodule. A lensless system is capable of magnification due to the projection of the image over a distance, and that diffraction widens the image with airy disks. The images are not focused due to diffraction and the non-coherence of the wave. Computer enhancement software was used that performs reverse diffraction [21]. The diffraction patterns recorded by the CMOS image sensor were reverse diffracted using the angular spectrum method (ASM). The program takes into account the diffraction distance, wavelength of light, and pixel size. For the lensless case, the resolution was improved from 23 lines per mm to 29 Lp/mm [20]. The imaging setup had a field of view of 6.14 mm × 4.604 mm and had a small pixel size (1.25 µm) that enabled digital magnification of small objects, such as small microbeads or cells.

### 2.8. Process for Imaging and Counting Cells

The CD4^+^ T cells isolated from a blood sample can be counted manually. A visual examination is needed for the identification of CD4^+^ T cells from unattached beads and debris. If the sample can be sufficiently purified to mainly CD4^+^ T cells and attached Dynabeads, then automatic counting based on the area can be employed to distinguish cells from one another. The ImageJ [22,23] program was customized to automatically count cells that were photographed on an inverted microscope (Nikon Eclipse TE2000-S) at 10×. The Appendix A provides details of the process of using ImageJ for various objects and conditions. The parameters were also optimized for images from the Fluorescent Cell Imager (ZOE, Bio-Rad Laboratories, Hercules, CA, USA) with 20× objective lens and for the images from lensless systems. The parameters are based on the type of object being imaged. We were able to distinguish CD4^+^ T cells and the beads that were not attached to cells. Dapi or NucBlue stains were used for fluorescence verification.

The Appendix A titled “Process for imaging and Counting Cells in ImageJ” and Appendix A show the basic ImageJ process parameters for counting the cells.

## 3. Results

### 3.1. Capture of CD4^+^ T Cells and Monocytes Using Dynabeads

Dynabeads functionalized with anti-human CD4 antibody (ThermoFisher, part number 11145D) [24] were utilized for the positive isolation of CD4^+^ T cells from whole blood samples. EDTA was added to the fresh blood as an anticoagulant. Moreover, ultrapure DPBS was used for washing purposes. The CD4 glycoprotein is expressed on two types of cells, i.e., CD4^+^ T lymphocytes and CD4^+^ monocytes. However, CD4^+^ T lymphocytes also express CD3 glycoprotein, whereas CD4^+^ monocytes do not express this CD3 antigen. Therefore, we used this biomarker to isolate the pure population of CD4^+^ T lymphocytes by binding them to the substrate and washing out the monocytes.

### 3.2. Microfluidic Device Process Verification

The CD4^+^ T cell isolation process was first verified by the standard method using an inverted microscope (Nikon Eclipse TE2000-S), a hemocytometer, and a Fluorescent Cell Imager (ZOE, Bio-Rad Laboratories, Hercules, CA, USA) microscope. Figure 4 presents the image of the cells obtained with the Fluorescent Cell Imager (ZOE, Bio-Rad Laboratories, Hercules, CA, USA). The lensless system that was developed could then be accurately utilized to the count the CD4^+^ T cells from the blood samples. The gold standard used for comparison was the standard Dynabeads magnetic separation with a hemocytometer count (HC) on an inverted microscope (Nikon Eclipse TE2000-S). Figure 5 portrays the Dynabeads and Dynabeads with attached CD4^+^ T cells on a section of the hemocytometer.

### 3.3. Process Verification Results with Blood Samples

The Continental Blood Bank, located locally in Fort Lauderdale, Fl., was the source of several different whole blood samples from which we were able to successfully isolate, identify, and count the CD4^+^ T cells using a microchip module. A wide-field optical setup and an automatic counting method were used to finalize the count. Three different samples are shown in Table 1. These are the whole blood samples. Sample 3 was first processed with the EasySep CD4^+^ T isolation kit to isolate the CD4^+^ T cells. Table 1 exhibits the results of the three different samples for specificity. Figure 4 portrays the CD4^+^ T cells attached to beads in a merged view of fluorescence and bright-field images that were taken on a Fluorescent Cell Imager (ZOE, Bio-Rad Laboratories, Hercules, CA). The image was enlarged to show the details of various CD4^+^ T cells containing beads and CD4^+^ T cells that are also attached to the substrate.

The microfluidic device process was proven to be effective. Table 1 portrays the efficiency of the process when comparing the microchip-based CD4^+^ T cell count to the hemocytometer count when using the Nikon as the imaging device. Multiple images are necessary for both the hemocytometer and the microchip module to ensure a sufficient sample. This is a standard procedure for the hemocytometer, and the cells were visually identified and averaged. We were able to take a string of images (eight consecutive images) down the center of the microchip module and form a unified image. Given a known volume and area of the module, the concentration could be calculated. The small field of view offered by a regular optical microscope demonstrated the need to create a wide-field imaging system. We develop these simplified systems as presented in the wide-field imaging section.

Table 2 shows the results of three different blood samples and their specificity. The same image was taken as a fluorescent image and then a bright-field image. The Fluorescent Cell Imager (ZOE, Bio-Rad Laboratories, Hercules, CA, USA) merged view shows a good correlation between the bright field and the blue field. NucBlue was used as the UV dye because it works with both alive and dead cells. It is anticipated that the final field process will be dealing primarily with live cells since the assay time will be short.

The microchip process has been proven for cell isolation efficiency (98.3 ± 10.8%) and specificity (89.3 ± 9.3%), as shown in Table 1 and Table 2, respectively. The next step is to show the results when using a lensless or wide-field lens system. The cell isolation efficiency was 88.5 ± 9.2%, as shown in Table 3. The specificity was already proven with the fluorescent imaging system.

### 3.4. Validation of Module Process and Counting Using Wide-Field Imaging

The microfluidic chip modules integrated with a lensless imaging system were designed to have a wide field of view so that the large surface area of the chip can be imaged rapidly. In CD4^+^ T cell isolation experiments, the number of cells to be counted per unit area is small. It is necessary to have either multiple images or a wide-area imaging system to capture a sufficient sample within the channel of the module.

Our previous work [20] has shown that wide-field lensless and custom-designed lens systems are possible. A low-cost lensless system is also described in that paper. For this work, we used the optical systems we had previously developed. A lensless system has a wide field of view, conforming to the size of the imaging field-of-view, in this case, 6.14 × 4.6 mm, as shown in Figure 3. The resolution is limited by the size of the pixel (1.25 um in this case) and the image area by the size of the array, 18 Megapixels. Figure 3 also portrays an enlarged section of the microchip module and shows the beads and CD4^+^ T cells. Table 3 shows that the CD4^+^ T cell isolation efficiency was at 89 ± 9.19 for the lensless system.

### 3.5. Conclusion and Discussion

A microfluidic chip-based assay has been developed that can sort out CD4^+^ T lymphocytes and count them using a microfluidic device, rather than magnetic separation, with excellent specificity and efficiency. The developed process was proven to be successful and provides 89.3 ± 9.3% capture specificity and 98.3 ± 10.8% capture efficiency inside microfluidic channels. The processes have been simplified such that a portable microchip design is possible for point-of-care use.

A wide-field optical system was explored, and its properties and limitations were examined. Lensless equipment was designed, built, and tested to complement the microchip design. This imaging system allows a larger sample area to enable the accurate counting and recognition of biological samples that normally have a low count per unit volume but at a lower cost per assay. The ImageJ-based algorithm was structured and used to automatically count cells. In the future, artificial intelligence could be used to better identify different types of cells.

Simple systems, such as lensless or wide-field lens systems, can be used at the point-of-care locations. This microfluidic device will enable wide-field optical systems to capture a sufficient area to count a low number of cell assays. Overall, the developed method is cost-effective, easy to use, and rapid, and can be used for counting CD4^+^ T cells at the point-of-care settings.

## Figures and Tables

**Figure 1 biosensors-12-00012-f001:**
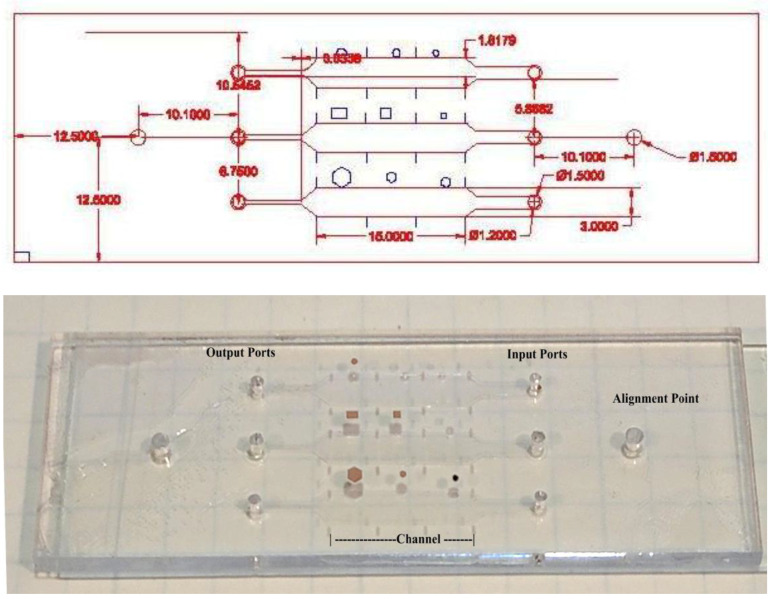
Microchip module with dimensions. The dimensions are in mm. The module is the size of a typical slide, 25 × 75 mm. The channel depth is 0.076 mm.

**Figure 2 biosensors-12-00012-f002:**
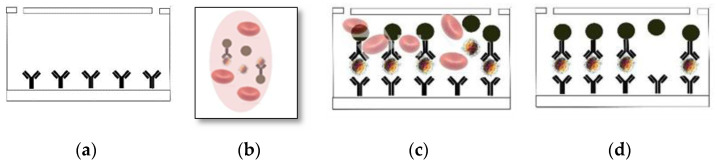
Basic functionalization process: (**a**) prepare substrate with Biotin anti-CD3 antibody; (**b**) mix blood with Dynabeads functionalized with anti-CD4 antibodies; (**c**) wash module with 400 μL DBPS at 50 μL/min; (**d**) observe and count cells.

**Figure 3 biosensors-12-00012-f003:**
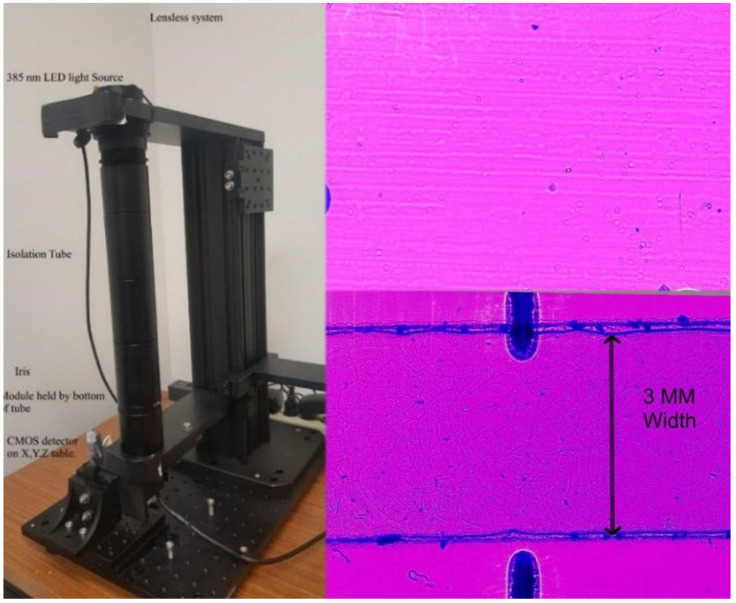
The left image is a lensless imaging system. LED is fitted inside the housing connected to isolation tube followed by 100 microns-sized pinhole. The light passes through the pinhole and shadow patterns of beads/cells are produced on CMOS sensor mounted on the base part. The chip containing captured cells and beads is placed on CMOS sensor before taking the image. The upper right lensless image was enlarged and shows beads and cells in a microfluidic chip module. The bottom right image shows the typical width of field, which is that of the imaging device.

**Figure 4 biosensors-12-00012-f004:**
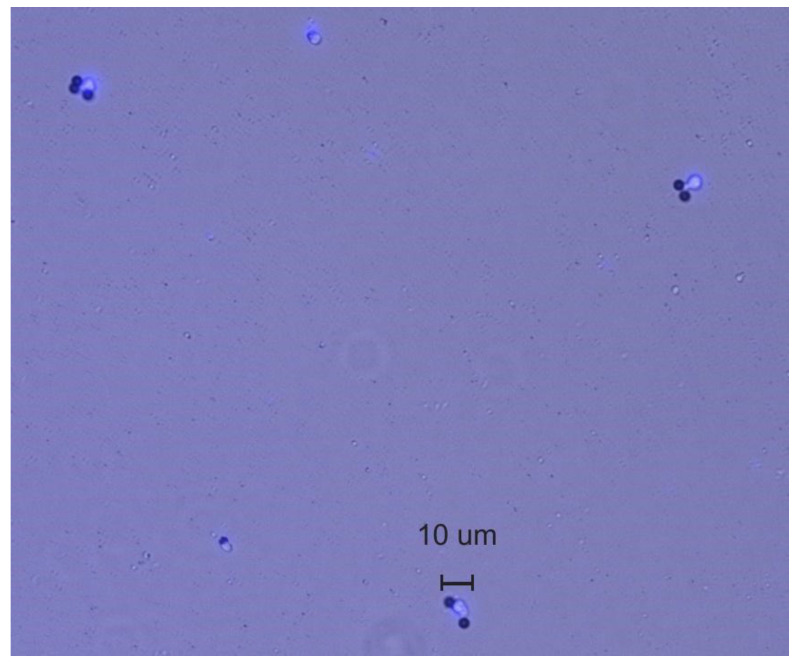
Enlarged merged image from Fluorescent Cell Imager (ZOE, Bio-Rad Laboratories, Hercules, CA, USA) showing beads and white cells.

**Figure 5 biosensors-12-00012-f005:**
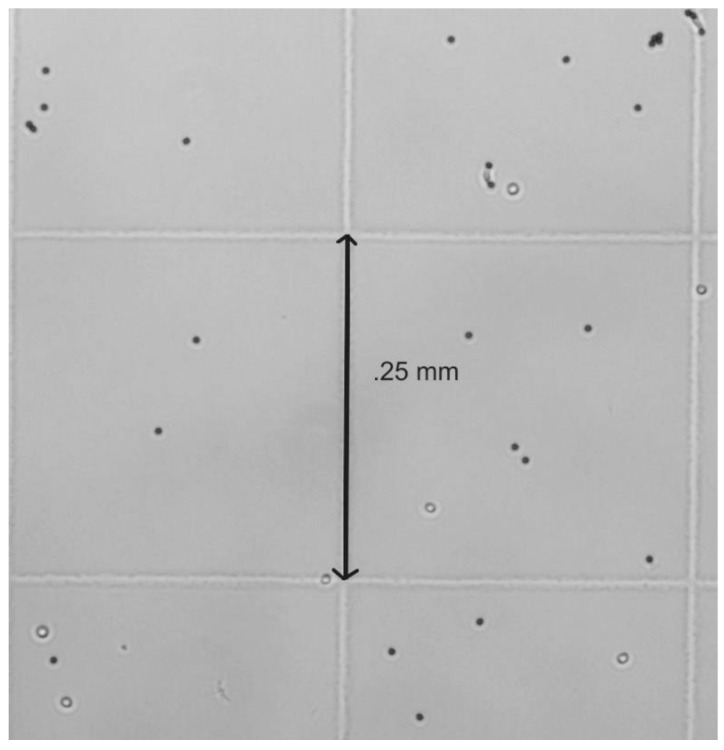
Hemocytometer showing CD4^+^ T cells and beads from standard Dynabeads process. Note that the normal Dynabeads process does not remove unattached beads since the magnetic force cannot distinguish the difference. Visual identification is required for area calculations.

**Table 1 biosensors-12-00012-t001:** Process efficiency of the micromodule based on inverted microscope (Nikon Eclipse TE2000-S) measurements.

Sample	HC Hemocytometer Count	MC Cell Count	Efficiency %
Sample 1 Blood	214	178	86
Sample 2 Blood	380	392	103
Sample 3 Blood (Processed with EasySep kit)	1592	1680	106
Average			98.3 ± 10.8

**Table 2 biosensors-12-00012-t002:** Fluorescent Cell Imager (ZOE, Bio-Rad Laboratories, Hercules, CA, USA) used for specificity measurement. Cells/image.

Sample	FC Blue Count	BF Bright Field	Specificity %
Sample 4 Blood	345	414	83
Sample 5 Blood	380	449	85
Sample 6 Blood (Processed with EasySep kit)	311	311	100
Average			89.3 ± 9.3

**Table 3 biosensors-12-00012-t003:** Lensless cell count efficiency using a lensless or lens optical system.

Sample Blood	Hemocytometer Count (HC)	Lensless Cell Count	Percentage Efficiency
Sample 7 Blood	380	311	82
Sample 8 Blood	480	457	95
Average			88.5 ± 9.2

## Data Availability

Not applicable.

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
