# Peer review of "Development of a Microfluidic Device for CD4+ T Cell Isolation and Automated Enumeration from Whole Blood"

_biosensors, 2021, doi:10.3390/bios12010012_

Round 1
Reviewer 1 Report
The authors present a microfluidics chip design for the counting of CD4+ T cells using a lensless imaging system which was previously developed by the same lab. They count the number of T cells captured by the UCHT1-anti-CD3 antibody and compare their results with measurements using a Hemocytometer, and calculate efficiency and specificity. The results confirm that the microfluidic chip could be used to replace hemocytometer counting. However, the manuscript is written in a very sloppy style with missing punctuation marks and parts of text, references to non-existing of wrong figures and chapters and inapprehensible phrases.
E.g.:
Line 60: This sentence comes without any context. I guess there is something missing before or it was copied to the wrong section.
Line 207: Where is the design of the wide area imaging technique described? Section III B does not exist. Do the authors mean chapter 2.9.?
Line 225: “.” Is missing
Line 240: There is no figure 8.
Line 243: the sentence is not finished
Line 244: What is a “reverse diffraction”. Do the authors mean deconvolution?
Line 313: What means “F A”?
Different fonts sizes in the supplementary information, wrong formatting of equations,…
General remark: Search for “fluorescent” in the document and substitute most of the terms with “fluorescence”
Author Response
We wish to thank reviewer for taking the time and effort to read and comment on the paper. We have addressed all the comments that significantly improved the paper. Please see the attached file for detailed response.

Reviewer 2 Report
The manuscript submitted by Fennell et al described a microfluidic device for CD4+ cell isolation and quantification from whole blood samples. This manuscript needs significant improvement before it can be further considered for publication.
- There have been a lot of microfluidic-based methods for CD4+ isolation and enumeration, the authors should add more references and provide a more comprehensive analysis of previous work. The authors should also compare their proposed approach with the existing methods and clearly present the advantages and novelty.
- It appears to the reviewer that results and methods were mixed up in “materials and methods”, and it is difficult to follow. The reviewer recommends the authors restructure the “materials and methods” section.
- Page 4, Figure 2 misses the caption of 2c and 2d. And the legend should be labeled.
- Page 6, line 176, could the author provide more details on how to control the washing step, since it is critical to ensure reproducibility and specificity. Different flow rates would significantly impact the final results.
- Page 6, Figure 5, it is not clear what the author would like to demonstrate in the purple color image, more clarification or labeling is needed.
- If the instrument is novel, and significant to the proposed work, it needs a more detailed description.
- “this second washing step ensures that only CD4+ cells and beads remain in the tube”. Need experimental support and re-structure the sentence.
- Line 193, microliter is not “uL”
- Figure 6, how many repeats? Is the error bar standard deviation?
- Figure 7, same issue as Figure 6.
- The average of the efficiency and the specificity is misleading. The deviation of efficiency is quite large, while it is not mentioned in the conclusion. For both of them, more samples should be conducted to give convincible results.
- Last but not least, although this device is portable, the microscope imager is not, and it still requires a number of manual operations, how can it be feasible for point of care? More clarification is required.
Author Response
We wish to thank the reviewer for taking the time and effort to read and comment on the paper. We have addressed all the comments that significantly improved the paper. Please see the attached file for the detailed response.

Reviewer 3 Report
This paper describes a detection system for CD3/4+ T cells based on microfluidics. The detection principle is manual/automatic counting of immobilized cells on the channel by biotinylated anti-CD3 antibody and the labeling with anti-CD4 magnetic beads.
First of all, what I notice in this paper is the severe problems in the scholarly presentation. First, too many abbreviations are used without explanation. All the abbreviations should be fully spelled and explained at its first appearance. Exceptionally, DPBS and DSA are explained in line 103 and line 107, respectively, but almost all the others are not adequately explained, such as HC in line 178 and line 208. I strongly advise to add a list of abbreviations at the end of manuscript for non-specialist readers.
Also, the way to introduce some reagents and apparatuses is so strange. The microscopes should not be called Nikon or ZOE (Biorad?), but inverted microscope (Nikon Model xx) or fluorescence cell imager (ZOE, Biorad, city, state or country) with 20x objective lens etc.
Some of essential information are not described in the main text. For example, the place and the way to immobilize biotinylated CD3 antibody should be clearly written. The reader cannot understand the reason of biotinylation and the place where the antibody is immobilized in Figure 1. Also, the way of calling the antibody should be unified. For example, CD4+T antibodies in line 134 should be written as anti-CD4 antibodies. Also, CD4+T cells should be written as CD4+ T cells.
It is also strange that many results are shown as Figures 3-6, Table 1 in not Results but in Materials and Methods section. In addition, the values shown in Figure 6 are the same as in Table 1 and 2, which is duplicate presentation and never allowed. In addition, the meaning of error bar is not described. It is also worth noting that one cannot average and calculate errors for the results obtained with different conditions. The same applies to Figure 7 and Table 3. In this case, it is not allowed to draw error bars using only two data. Additional data should be taken, and Figures 6 and 7 should be removed.
Figure 5 is not easy to understand. There should be tangible scheme of the lensless lab imaging system. It is not easy to understand the system just from the current pictures.
Abstract and line 132, the word “affinity” is not an appropriate word to explain increased binding kinetics. The word “avidity” should be used instead. Also because it is not 1:1 interaction, Ka value should not be used for the explanation.
Line 266, where is the Continental Blood Bank?
ul and um should be re-written as µL and µm, respectively.
In overall, I conclude that this manuscript has not been designed and written with sufficient care.
Author Response
We wish to thank the reviewer for taking the time and effort to read and comment on the paper. We have addressed all the comments that significantly improved the paper. Please see the attached file.

Round 2
Reviewer 2 Report
The authors have properly addressed the reviewer's comments. The manuscript meets the standard and can be considered for publication.
Author Response
Attached please find the response to Reviewer's comments.

Reviewer 3 Report
In this revised manuscript, although some of my comments were answered as expected, many of them remain unanswered adequately. For example, I requested to add List of abbreviations at the end of main text, which is the usual place, they put the list in SI, which was not alphabetically ordered.
Also, although I advised to revise the presentations of the Nikon and ZOE machines appropriately (“The microscopes should not be called Nikon or ZOE, but inverted microscope (Nikon Model xx) or fluorescence cell imager (ZOE, Bio-Rad, city, state or country) with 20x objective lens etc.“), they did not.
Also I am saying that the locations of the Tables should be in Results, not in Materials and Methods, but their answer was “The Tables and Figures were intentionally put at the end of the writing to make it easier to finally place them when printing to a specific journal”, which does not make sense.
More importantly, the Tables should have correct average and SD values as follow, considering the valid digits.
In Table 1, 98±10.78 % should be 98.3 ± 10.8 %
In Table 2, 89±9.29 % should be 89.3 ± 9.3%
In Table 3, 89±9.19 % should be 88.5% ± 9.2 %
I still have to say that in the former two tables, the samples used were not equally treated. In Table 3, only two values were still used for the calculation, which is not statistically recommended.
Line 150, “The developed assay utilizes a microfluidic chip coated with an improved antibody avidity from the CD3 antibody. The supplement II D. has more information on stronger avidity binding.” does not make sense. They should be re-written as
“The developed assay utilizes the stronger avidity of anti-CD3 antibody coated on a microfluidic chip. More explanation on stronger avidity is given in the supplement IID.”
Unless these flaws are to be corrected, I will never recommend this manuscript to be accepted.
Author Response

(The authors gave the same response as above.)
